# Search for scalar induced gravitational waves in the international pulsar timing array data release 2 and NANOgrav 12.5 years datasets

Virgile Dandoy[1*], Valerie Domcke[2†] and Fabrizio Rompineve[2,3,4‡]

**1** Institut für Astroteilchenphysik, Karlsruhe Institute of Technology (KIT),
Hermann-von-Helmholtz-Platz 1, 76344 Eggenstein-Leopoldshafen, Germany
**2** CERN, Theoretical Physics Department, 1211 Geneva 23, Switzerland
**3** Departament de Física, Universitat Autònoma de Barcelona,
08193 Bellaterra, Barcelona, Spain
**4** Institut de Física d'Altes Energies (IFAE) and The Barcelona Institute of Science
and Technology (BIST), Campus UAB, 08193 Bellaterra (Barcelona), Spain

⋆ virgile.dandoy@kit.edu , † valerie.domcke@cern.ch , ‡ fabrizio.rompineve@cern.ch

## Abstract

We perform a Bayesian search in the latest Pulsar Timing Array (PTA) datasets for a stochastic gravitational wave (GW) background sourced by curvature perturbations at scales $10^5 \text{ Mpc}^{-1} \lesssim k \lesssim 10^8 \text{ Mpc}^{-1}$. These re-enter the Hubble horizon at temperatures around and below the QCD crossover phase transition in the early Universe. We include a stochastic background of astrophysical origin in our search and properly account for constraints on the curvature power spectrum from the overproduction of primordial black holes (PBHs). We find that the International PTA Data Release 2 significantly favors the astrophysical model for its reported common-spectrum process, over the curvature-induced background. On the other hand, the two interpretations fit the NANOgrav 12.5 years dataset equally well. We then set new upper limits on the amplitude of the curvature power spectrum at small scales. These are independent from, and competitive with, indirect astrophysical bounds from the abundance of PBH dark matter. Upcoming PTA data releases will provide the strongest probe of the curvature power spectrum around the QCD epoch.

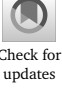

# 1  Introduction

The detection of the stochastic background of Gravitational Waves (GWs) is one of the primary targets of current [1–4] and future (e.g. [5–7]) GW observatories. Any sufficiently violent process occurring in the Universe, no matter how early in the cosmological history, would contribute to such a background, which then holds the promise of offering a new probe of fundamental physics before the epoch of recombination and possibly at very high energy scales.

Recently, all currently active Pulsar Timing Array (PTA) observatories (NANOgrav [2], Parkes PTA [3], European PTA [4], and their joint effort International PTA [8]) have claimed strong evidence for a common-spectrum process in their datasets, at frequencies $(1-10)$ nHz. Upcoming and near-future data releases from the same PTAs are expected to have enough sensitivity to draw conclusions on the nature of such a process [9], in particular whether it exhibits the characteristic tensorial ("Hellings-Downs") correlations [10] of a GW background. A signal at these frequencies is expected from mergers of supermassive black hole binaries (SMBHBs), although its amplitude and spectral properties are currently not uniquely predicted by astrophysical models (see e.g. [11,12]). If evidence for quadrupolar correlation is found (not necessarily related to the current excess), it will be crucial to understand if the GW signal contains any significant contribution of cosmological origin. Detailed studies of GW spectra from cosmological phenomena, as well as searches for such signals, are then required to properly interpret PTA data (see [13–18] for recent work).

In this paper, we join such an effort by performing a Bayesian search for GWs radiated by scalar (curvature) perturbations in the early Universe, focusing on the NANOgrav 12.5 years [19] (NG12) and International PTA Data Release 2 [20] (IPTA DR2) datasets (other recent datasets have been shown to give similar information, see [8]). Such a background of GWs is distinct from the tensor modes generated by de Sitter fluctuations during inflation together with scalar perturbations. In the perturbative expansion of the metric according to General Relativity (GR), scalar and tensor modes do not mix at first order, but second-order tensor perturbations are sourced by first order scalar modes [21–26] (see also [27–29] for recent proof of gauge independence). Therefore, scalar induced GWs are a general consequence of GR and of the very existence of structures in the Universe. However, only large enough scalar perturbations lead to an observable GW background. In practice, observations of the Cosmic Microwave Background (CMB) as well as of the power spectrum of Large Scale

Structure (LSS) constrain the amplitude of the curvature power spectrum to be $P_\zeta \sim 10^{-9}$ and almost scale-invariant for $k \lesssim \text{Mpc}^{-1}$, which then implies that only a very small amount of GWs is produced at those large scales. On the other hand, the properties of the power spectrum at smaller scales are unknown to a large extent. For instance, deviations from approximate scale invariance may in principle occur and the spectrum might be significantly enhanced compared to CMB scales. This possibility is in fact often invoked as a mechanism to form Primordial Black Holes (PBHs) [30, 31] from the collapse of such large density perturbations. The frequency range of PTAs correspond to scales $k \sim (10^6 - 10^8) \text{Mpc}^{-1}$, which entered the Hubble horizon around and below the epoch of the QCD crossover in the hot Universe. Therefore, large deviations from scale invariance at such scales can be indirectly probed by PTAs, via the induced GW signal [32, 33] (see also [34–36]). Additionally, as mentioned above, the same large perturbations that source GWs may also lead to a significant fraction of PBHs, with masses $\sim (0.001 - 1000)M_\odot$, encompassing the binary BH mass range currently observed at the LIGO/Virgo/KAGRA interferometers. Therefore, PTAs can potentially play an interesting role in detecting signatures of PBH dark matter (in fact, currently the strongest constraints on the power spectrum at PTA scales are indirectly derived from bounds on the fraction of dark matter (DM) in PBHs of a given mass, see e.g. [37–39], see also [40] for recent progress on setting constraints from the formation of dark matter minihalos).

While some studies have recently appeared with similar focus (see [35, 41] for searches in the NG11 and NG12 datasets respectively, and [13, 42–45] for interpretations of the NG12 excess, see also [46] for a search in the LIGO/Virgo/KAGRA O3 data [1], which probe much smaller scales than PTAs), our work presents important novelties. First, we perform a Bayesian search in the IPTA DR2 dataset (in addition to NG12, where differently from [41] we use only the first five frequency bins following the NG12 collaboration [2]). The IPTA DR2 dataset prefers a larger amplitude of the stochastic GW background compared to NG12, as well as a different spectral slope (although the two datasets are in less than $3\sigma$ tension assuming a power-law model [8]), therefore our search provides new insight into the interpretation of the common-spectrum process in terms of scalar induced GWs. Second, we include the expected stochastic GW background of astrophysical origin in our searches. Third, we pay close attention to constraints on the amplitude of the power spectrum from the overproduction of PBHs, which we re-assess highlighting the corresponding theoretical uncertainties and clarifying existing claims in the literature. Overall, these novelties allow us to properly assess the likelihood of the scalar induced GW interpretation over the astrophysical model. As a result, we also derive upper limits on the amplitude of the power spectrum at small scales, in the presence of a SMBHBs-like common-spectrum signal. These constraints are independent from indirect astrophysical bounds on PBHs. Throughout this work, we make use of a log-normal parametrization for the scalar power spectrum at the scales probed by PTAs, which allows us to investigate both spectra with a broad or a narrow peak in the PTA frequency range.

This paper is structured as follows: in Sec. 2 we review the GW spectrum from scalar perturbations; in Sec. 3 we present constraints on the power spectrum from the overproduction of PBHs; Sec. 4 is devoted to the results of our searches, which we relate to previous work and claims in Sec. 5. Finally, we conclude in Sec. 6. This paper also contains three Appendices, where we review details of: A the scalar induced GW spectrum; B the computation of the PBH abundance to derive constraints on the power spectrum; C our numerical strategy in the search.

## 2 Gravitational waves from scalar perturbations

The spectrum of GWs generated at second order by scalar perturbations depends in general on the amplitude and shape of the curvature power spectrum. In the inflationary paradigm, the latter is set by the inflaton dynamics (thus by the shape of the inflaton potential) roughly when $\simeq \log_{10}[k/(0.05 \text{ Mpc}^{-1})]$ e-folds have passed since the generation of CMB anisotropies. Here $k$ is the wavenumber of a given curvature mode today, related to the characteristic frequency $f$ of the induced GWs by

$$ f = \frac{k}{2\pi} \simeq 1.6 \cdot 10^{-9} \text{ Hz} \left( \frac{k}{10^6 \text{ Mpc}^{-1}} \right). \tag{1} $$

For PTA searches we are interested in scales $k \sim (10^5 - 10^8) \text{ Mpc}^{-1}$ which exited the inflationary Hubble sphere $\simeq 15 - 20$ efolds after the CMB pivot scale. Beside constraints from $\mu$-distortions for $k \lesssim 5 \cdot 10^5 \text{ Mpc}^{-1}$ [47,48], we do not have any direct probe of the curvature power spectrum at those scales.

The detailed feature of the inflaton potential and dynamics in a given model will then set the amplitude and shape of the resulting curvature power spectrum and GW signal, which are thus necessarily model-dependent. In fact, a significant amount of GWs can be produced only when the amplitude at small scales is much larger than at CMB scales. For this to be possible, the inflaton dynamics should exhibit some strong deviations from scale invariance, which may then lead to an enhancement in the curvature power spectrum. To the aim of remaining agnostic about the specifics of the inflationary model, in this work we shall make use of a simple log-normal parametrization that captures the possibility of a peak in the power spectrum at small scales:

$$ P_\zeta(k) = \frac{A_\zeta}{\sqrt{2\pi}\Delta} \exp\left[ -\frac{\log^2(k/k_\star)}{2\Delta^2} \right], \tag{2} $$

where $k_\star$ is the peak scale and $\sim \Delta$ the width. Following conventions, the normalization of the spectrum is such that $A_\zeta$ represents the amplitude of the integrated power spectrum (over $k$), rather than the peak amplitude $A_\zeta/(\sqrt{2\pi}\Delta)$. For $\Delta \sim 1$, the peak is broad and for large $\Delta$ it is essentially flat over a large range of wavenumbers. On the other hand, for $\Delta \ll 1$ the peak is narrow, and it reduces to a Dirac delta function $P_\zeta(k) = A_\zeta \delta[\log(k/k_\star)]$ as $\Delta \to 0$. While our analysis assumes a power spectrum of the form (2), we expect our results to apply qualitatively to other peaked and broad spectra as well.

As for any cosmological source, the stochastic GW background radiated by scalar perturbations is typically expressed in terms of its relic abundance $\Omega_{\text{gw}}(f) \equiv d\rho_{\text{gw}}^0 / d\log(f)/(3H_0^2 M_p^2)$, where $f$ is the frequency of the GWs, the superscript $0$ means that the energy density in GWs should be evaluated at present times, $H_0$ is the Hubble expansion rate today and $M_p \equiv (8\pi G_N)^{-1/2}$ is the reduced Planck mass. The calculation of $\Omega_{\text{gw}}(f)$ for the case of interest corresponds to computing the four-point function of scalar modes [27–29,50,51]. The result can be expressed in the following compact form, (throughout this work, we assume a radiation dominated Universe at the time of re-entry of the perturbations of interest and until matter-radiation equality)

$$ \Omega_{\text{gw}} h^2 \simeq 1.9 \cdot 10^{-9} \left( \frac{A_\zeta}{0.01} \right)^2 \left( \frac{g_*(T_\star)}{17.25} \right) \left( \frac{g_{*s}(T_\star)}{17.25} \right)^{-\frac{4}{3}} \left( \frac{\Omega_r^0 h^2}{2.6 \times 10^{-5}} \right) S\left( \frac{f}{f_\star}, \Delta \right), \tag{3} $$

where $g_*(T_\star)$ is the number of relativistic degrees of freedom in the early Universe at the temperature $T_\star$ when the mode $k_\star$ re-enters the horizon, defined by $k_\star = aH_\star$, see Fig. 2 (the reference value in the equation above corresponds to re-entry slightly below the QCD crossover

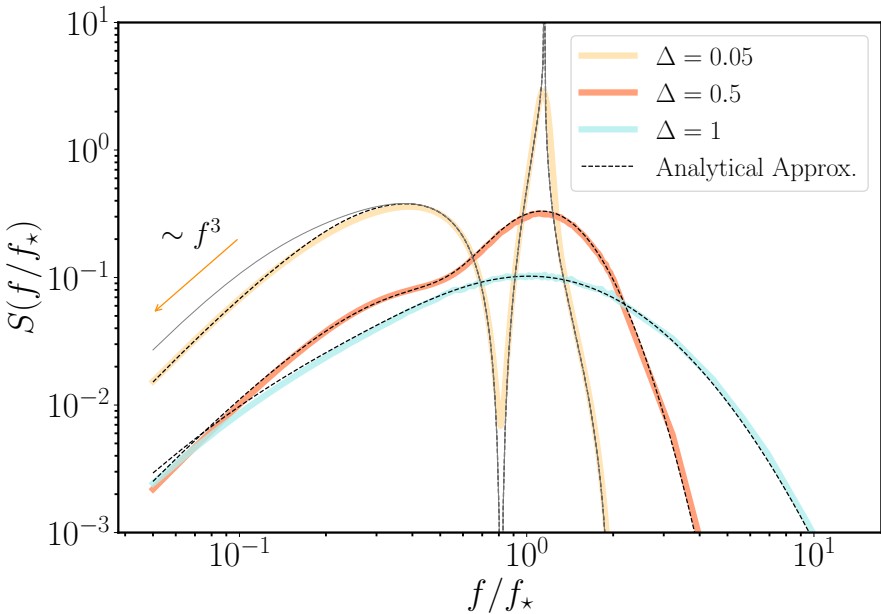

Figure 1: Spectral shape of the GW signal induced by scalar perturbations with log-normal power spectrum. The thick curves are obtained by numerical evaluation, see App. A, for different values of $\Delta$. The analytical approximations are shown as black dashed curves and coincide with the ones of [49] in the whole range of $f/f_\star$ for $\Delta = 0.5, 1$. For the narrow peak case $\Delta = 0.05$, the approximation of [49] is shown as the solid gray line and the deviation from the exact numerical value is clearly visible in the IR. Finally the $\sim f^3$ trend is shown in the IR by the orange arrow.

$T_\star \lesssim 100$ MeV), and $\Omega_r^0 h^2$ is the relic abundance of radiation today. The function $S(f/f_\star)$ encodes the spectral shape of the signal and can in general only be evaluated numerically (see App. A). We show it in Fig. 1 for representative values of $\Delta$. Its general features are nonetheless easy to understand: first, it is peaked close to the frequency $f_\star$ corresponding to the wavenumber $k_\star$. Second, it increases as $f^3$ for $f \ll f_\star$, as dictated by causality. Third, it exponentially decays for $f \gg f_\star$, following the decrease of the curvature power spectrum.[1] The precise location of the peak and the behavior around it depends on the width of the power spectrum $\Delta$ [49]. For $\Delta \gtrsim 1$, there is a log-normal peak at $f = f_\star$, with a width $\Delta/\sqrt{2}$. On the other hand, for $\Delta \lesssim 0.2$, a two-peak structure appears [52]: a sharper peak at $f/f_\star = (2/\sqrt{3})e^{-\Delta^2}$ and a broader one at $f/f_\star = 1/e$, separated by a dip at $f/f_\star = (\sqrt{2/3})e^{-\Delta^2}$. The presence of the sharp peak is due to resonant amplification of tensor modes, see [52]. Notice that as $\Delta \to 0$, the amplitude of the IR tail of the GW signal, arising from scalar perturbations with anti-aligned wave vectors generating an IR GW, is independent of $\Delta$. We note that in this case, there is an intermediate region with slope $f^2$ for $f_\star > f \gtrsim 2\Delta e^{-\Delta^2}$. As $\Delta$ decreases, the causality tail can then be effectively shifted beyond the sensitivity band of PTAs.

An important limitation arises however when searching for narrow peaks in PTA datasets. The frequency resolution of these measurements is given by $\Delta f \simeq 1/T_{\text{obs}}$, where $T_{\text{obs}}$ is the longest observation timespan of a pulsar in the dataset. For NG12 (IPTA DR2), this is $T_{\text{obs}} = 12.5\,(29)$ years. Therefore, NG12 (IPTA DR2) cannot currently resolve peaks which are narrower than $\Delta f \simeq 2.5(1)$ nHz. A possible approach to deal with such GW spectra is

---

[1]For a peaked power-law curvature spectrum instead, the GW signal also decreases as a power law. To cover the possibility of a milder decrease at high frequencies in the PTA band using the log-normal spectrum, one can choose $\Delta > 1$ in (2).

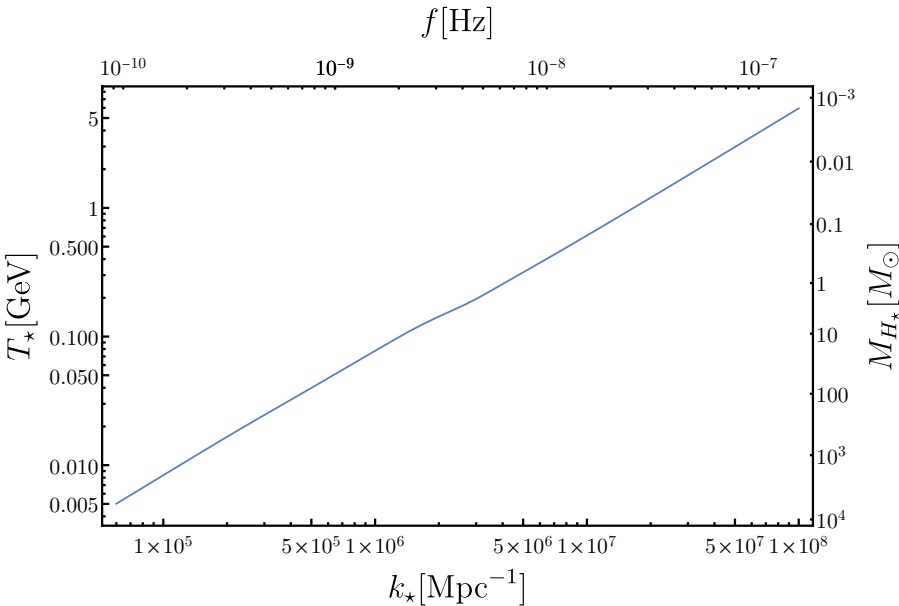

Figure 2: The temperature in the early Universe corresponding to horizon re-entry of the $k_\star$-mode. The slight deviation from a straight line around $k_\star \gtrsim 10^6$ Mpc$^{-1}$ is due to the rapid change of relativistic degrees of freedom during the QCD crossover. On the upper horizontal axis the characteristic frequency of the scalar induced GW signal is shown. On the right vertical axis, the horizon mass at the temperature $T_\star$ is shown.

to smoothen the peak region, for instance by averaging over the typical bin separation. This procedure however depends on the value of $\Delta$ and is computationally intensive. Alternatively, one can simply restrict the analysis to the IR tail of the signal (defined roughly by the frequencies smaller than the dip location). In this work, we choose this latter strategy to investigate spectra with $\Delta < 0.5$. From the point of view of setting constraints, this is clearly a conservative choice. We will comment further on the validity of our conclusions for narrow peaked spectra below.

Finally, let us discuss a technical point. The numerical calculation of (A.1) turns out to be rather slow, therefore significantly increasing the computational time required to explore the parameter space in our search. However, good approximations to the full numerical results have been obtained in [49]. We have further improved on these for the narrow peak case (see App. A for the explicit expressions). In our searches, we use these functions, plotted in Fig. 1 as dashed curves (notice the difference with the solid gray curve from [49] for $\Delta = 0.05$).

## 3 Cosmology

PTAs are currently sensitive to GW signals with $\Omega_{\text{gw}} h^2 \gtrsim 10^{-10}$ in the frequency band $f \sim 10^{-9} - 10^{-8}$ Hz. According to (3), (1) and Fig. 1, this corresponds to scalar spectra peaked at $k_\star \sim (10^5 - 10^8)$ Mpc$^{-1}$ (for $k_\star \sim 10^6 - 10^7$ Mpc$^{-1}$, the peak of the signal lies in the sensitivity band), with amplitudes $A_\zeta \gtrsim 0.01$. Upon horizon re-entry, such large perturbations may then undergo gravitational collapse and form primordial black holes (PBHs). These make up a fraction $f_{\text{PBH}} \equiv \Omega_{\text{PBH}}/\Omega_{\text{DM}}$ of the DM today. Larger amplitudes lead to larger values of $f_{\text{PBH}}$, therefore a limiting value of $A_\zeta$ exists above which PBHs overclose the Universe (for the power spectrum (2), this value is a function of $k_\star$ and $\Delta$). In searching for scalar induced

GWs, it is thus important to impose an upper bound on $A_\zeta$ such as to avoid exploring regions of parameter space that are in contradiction with cosmological observations. The aim of this section is thus to present such a constraint (see [46] for analogous constraints on scales relevant for LIGO searches) as well as to clarify some aspects of the existing literature related to bounds on $f_{\mathrm{PBH}}$ from PTAs. A short review and further details are provided in App. B.

The fraction of dark matter in PBHs today can be expressed as [53–56]

$$f_{\mathrm{PBH}} = \frac{1}{\Omega_{\mathrm{DM}}} \int d\log M \int d\log k \, \beta_k(M) \frac{\rho_\gamma(T_k)}{\rho_c^0} \frac{s^0}{s(T_k)}, \qquad (4)$$

in terms of $\beta_k$, that is the fraction of the radiation energy density that collapses to PBHs of mass $M$ at horizon re-entry, and $\rho_c^0 = 3H_0^2 M_p^2$. Above $s(T_k)$ and $s^0$ are the entropy density at the temperature $T_k$ defined by $k = aH(T_k)$ and today, respectively. In the Press-Schechter formalism for spherical collapse [55] (see also [54]) and assuming Gaussianly distributed perturbations [57] (we comment on the effects of non-Gaussianities for our analysis below) one has

$$\beta_k(M) = \int_{\delta_c}^{\infty} d\delta_l \, \frac{M(\delta_l)}{M_H(k)} \frac{\exp\left(-\frac{\delta_l^2}{2\sigma(k)^2}\right)}{\sqrt{2\pi}\sigma(k)} \delta_D\left[\ln\frac{M}{M(\delta_l)}\right]. \qquad (5)$$

Here $\delta_l$ is defined by $\delta_l \sim -k\zeta'(k)$, where $\zeta$ is the curvature perturbation and $\delta_c \sim \mathcal{O}(1/3)$ is the critical threshold for gravitational collapse of a density perturbation during radiation domination.[2] At the linear level, $\delta_l$ coincides with the total matter density contrast, i.e. $\delta_l = \delta_m \equiv \delta\rho_m/\rho$ (we use instead the full non-linear relation, appropriate for large perturbations considered in this work [58–60]; its impact on PBH production is reviewed in App. B). The function $M(\delta_l)$ describes the actual mass of PBHs resulting from the collapse of the perturbation $\delta_l$, $M(\delta_l) = \kappa M_H(k)(\delta_m - \delta_c)^\gamma$, with $\gamma \approx 0.36$ [61,62] for a radiation dominated Universe and $\kappa \sim 1 - 10$ [58], and it is generically close to the horizon mass at re-entry

$$M_H(k) \equiv 4\pi \frac{M_p^2}{H} \simeq 20 \, M_\odot \left(\frac{k}{10^6 \, \mathrm{Mpc}^{-1}}\right)^{-2} \left[\frac{g_{*,s}^4(T_k)g_*^{-3}(T_k)}{17.25}\right]^{-1/6}, \qquad (6)$$

where we have normalized the $k$-scale to a typical value of interest for PTA searches, and consequently also normalized the number of relativistic species to its SM value around the corresponding horizon re-entry temperature $T_k \lesssim 100$ MeV (see also Fig. 2). The variance $\sigma$ of $\delta_l$ is computed as

$$\sigma^2(k) = \int_0^{\infty} \frac{dk'}{k'} W^2(k',k) P_\delta(k') = \frac{4}{9}\Phi^2 \int_0^{\infty} \frac{dk'}{k'} \left(\frac{k'}{k}\right)^4 T^2(k',k) W^2(k',k) P_\zeta(k'), \quad (7)$$

where $T^2$ is the linear transfer function, $W(k',k)$ is a so-called window function used to smooth the matter perturbations $\delta_m$ [55] (see also [63]) and $\Phi$ is a function that depends on the equation of state of the Universe (see App. B).

We can now make the following two remarks. First, the relic PBH abundance depends exponentially on the value of the critical threshold $\delta_c$, via the dependence on $M(\delta_l)$. This means that the relic abundance can be reliably computed only as long as the critical threshold can be accurately determined. The latter computation depends importantly on the shape of the power spectrum and can be performed analytically and numerically [64–67]. Second, there is an exponential dependence on the variance, whose computation relies on the

---

[2]The expression for $\beta_k(M)$ above differs from that obtained using peak theory (PT) rather than the Press-Schechter formalism, see e.g. [58]). We comment on the effects of using PT for the constraints presented in this work in App. B. We thank G. Franciolini, I. Musco, A. Urbano and P. Pani for pointing this out to us.

precise shape of the smoothing function $W$, for which there is currently no unique prescription [42, 56, 58, 63, 68]. Common choices in the literature include a real top-hat function $W = \left(3\sin(k/k') - k/k'\cos(k/k')\right)/(k/k')^3$ (this is just a step function in real space) and a (modified) Gaussian function $W = \exp(-(k/k')^2/4)$. Importantly, the actual value of the critical threshold also depends on this choice [64–67].

Overall, these two sources of delicate sensitivity currently prevent a reliable estimate of the relic PBH fraction. Indeed, for a given choice of curvature power spectrum parameters, the resulting PBH fraction may vary by more than five orders of magnitude depending on the choice of window function, even when the critical threshold is computed consistently (otherwise the change is typically much bigger). Therefore, we conclude that any attempt to derive constraints on (or evidence for given values of) $f_{\mathrm{PBH}}$ from PTA datasets, which indirectly probe the curvature power spectrum, is currently plagued by very large uncertainties [42, 56, 58, 63, 68].

On the other hand, the exponential sensitivity of $f_{\mathrm{PBH}}$ on the parameters of the power spectrum, in our case $A_\zeta$ in particular, also implies that the condition $f_{\mathrm{PBH}} \leq 1$ imposes an upper bound on $A_\zeta$ with only $O(1)$ uncertainties. We have derived such a bound for the two choices of window function discussed above, by solving the condition $f_{\mathrm{PBH}} = 1$ for fixed peak width $\Delta$ and for a set of values of $k_\star$. Our result is shown by the dotted lines in Fig. 6 (see also Fig. 3) and imposes $A_\zeta \lesssim 0.01 - 0.04$ (see below for specific choice of priors for our searches). The difference between the two curves corresponding to a different choice of window function can be considered as the theoretical uncertainty on the constraint (other common choices of window function give similar curves in between those reported in Fig. 6). It properly accounts for the non-linear relation between $\delta_m$ and $\delta_l$, and is based on a consistent choice of window function to compute the threshold as well as the variance. Additionally, we have included the effects of the change in the equation of state of the radiation background due to the occurence of the QCD crossover at the scales $k_\star \sim (10^6 - 10^7)\,\mathrm{Mpc}^{-1}$ (according to the recent analyses of [69, 70], we also included the results of [71] on the dependence of the prefactor between $\delta_l$ and $\zeta'(k)$), which lowers the critical threshold for collapse, thereby giving the visible dip in the $f_{\mathrm{PBH}} = 1$ curves.[3] The interested reader can find a detailed derivation in App. B. We notice that for fixed power spectrum parameters using a modified Gaussian window function always leads to a larger value of $f_{\mathrm{PBH}}$ than a top-hat function (a standard Gaussian gives a result in between these two, close to the modified Gaussian case).

Let us now briefly comment on non-gaussianities of the curvature perturbations. These are expected for such large perturbations as considered in this work and generically have the effect of lowering the critical threshold for collapse. A precise assessment of these effects is however still under investigation (see e.g. [74–78] for recent progress), therefore we assume Gaussianly distributed perturbations in the computation leading to the results shown in Figs. 6. We expect that their inclusion would shift the curves of $f_{\mathrm{PBH}} = 1$ to smaller values of $A_\zeta$, thereby making the overproduction constraint stronger. In this respect, our curve remains conservative. We notice that neglecting the non-linear relation between $\delta_m$ and $\delta_l$ has the same effect of raising the value of $f_{\mathrm{PBH}}$ corresponding to a given choice of power spectrum parameters. This is relevant for comparison of our results with previous literature (see Sec. 5). More details about the impact of different thresholds can be found in App. B.

---

[3]The impact of the QCD crossover on the scalar induced GW spectrum [72] (see also [73]) is small enough that we expect it not to significantly affect our search, given the current resolution of PTAs. Therefore, in our analysis of GWs we use a radiation background with a constant equation of state.

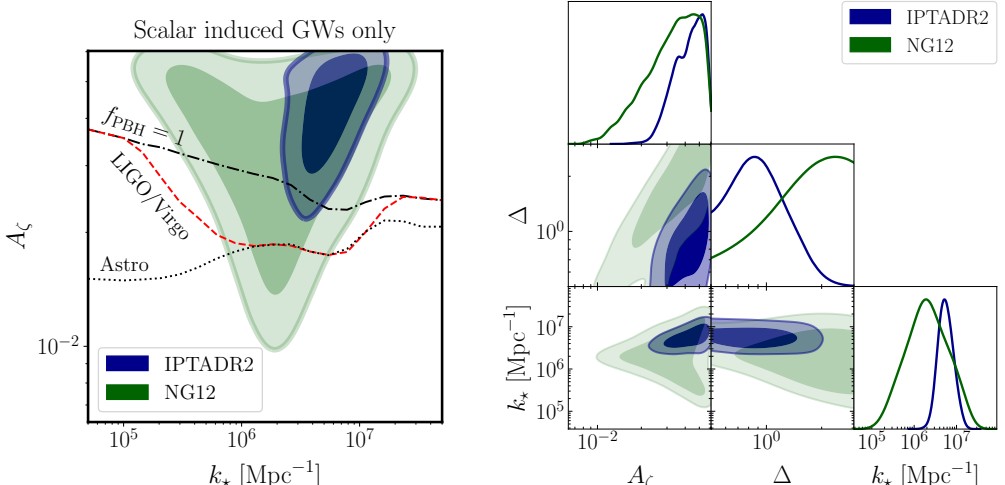

Figure 3: One- and two-dimensional posterior distributions for the parameters of the stochastic gravitational wave background sourced by curvature perturbations, assuming no other source of GWs is present. A conservative upper prior on $A_\zeta$ from overproduction of PBHs has been applied $\log_{10} A_\zeta \leq -1.22$, see text for details. The dark (light) shaded regions show 68% and 95% C.L. regions respectively. In the left panel, the region above the dot-dashed (dotted) black curve is constrained by PBH overproduction (astrophysical observations), for $\Delta = 1$. The region above the dashed red curve is constrained by LIGO/Virgo for $\Delta = 1$, see [79] and App. B. All the constraints in the plot are obtained using a top-hat window function. They would be stronger (weaker) for smaller (larger) $\Delta$.

## 4 Datasets and results

We now move to the presentation of our searches in PTA datasets for a stochastic GW background sourced by scalar perturbations. We make use of the publicly available NG12 and IPTA DR2 datasets. PTA searches are performed in terms of the timing-residual cross-power spectral density $S_{ab}(f) \equiv \Gamma_{ab} h_c^2(f)/(12\pi^2) f^{-3}$, where $h_c(f) \simeq 1.26 \cdot 10^{-18} (\text{Hz}/f) \sqrt{h^2 \Omega_{\text{GW}}(f)}$ (see e.g. [80]) is the characteristic strain spectrum and $\Gamma_{ab}$ is the Overlap Reduction Function (ORF) containing correlation coefficients between pulsars $a$ and $b$ in a given PTA.

We performed Bayesian analyses using the codes `enterprise` [81] and `enterprise_extensions` [82], in which we implemented the scalar induced GW signal (3) (with the approximations for the spectral shapes reported in App. A and restricting the search to the IR tail of the signal for $\Delta \leq 0.5$) and `PTMCMC` [83] to obtain MonteCarlo samples. We properly account for the temperature dependence of the number of relativistic degrees of freedom $g_*$ in the plasma (assuming SM degrees of freedom only), by implementing the results of [84]. This is relevant for the values of $k_*$ under consideration, since they encompass the QCD crossover where $g_*$ is most rapidly varying.

We derive posterior distributions and upper limits using `GetDist` [85]. We include white, red and dispersion measures noise parameters following the choices of the NG12 [2] and IP-TADR2 [8] searches for a common-spectrum process. Furthermore, we limit the stochastic GW search to the lowest 5 and 13 frequency bins of the NG12 and IPTADR2 datasets respectively to avoid pulsar-intrinsic excess noise at high frequencies, as in [2, 8]. More details about the numerical strategy, as well as full list of prior choices for our runs, are reported in App. C.

We start by performing detection analyses. That is, we look for the region of parameter space where a scalar induced GW background can provide a good model of PTA data. We

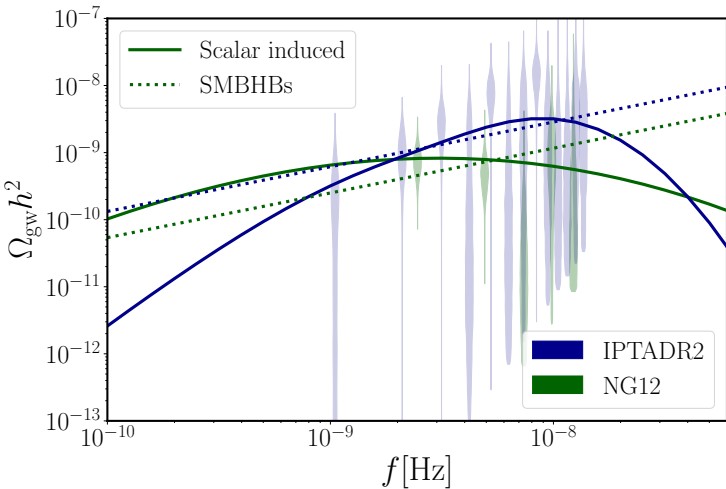

Figure 4: Relic GW spectra obtained for maximal likelihood values of parameters, as obtained from our searches. The solid curves show the scalar induced GW spectrum, according to our "Scalar induced GW only" search and are obtained setting $A_\zeta \simeq 0.04\,(0.04), k_\star \simeq 5.5\,(2.2)\cdot 10^6$ Mpc$^{-1}$, $\Delta \simeq 0.9\,(2.1)$ for IPTA DR2 (NG12). The dashed curves show the background from SMBHBs, according to our search for an astrophysical background only, and are obtained setting $A_{\text{SMBHBs}} \simeq 3\,(2)\cdot 10^{-15}$ for IPTA DR2 (NG12). The free spectrum posteriors obtained by converting the results of [2] (NG12) and [8] (IPTA DR2) are also shown (violin shapes, lower limits due to prior choices).

consider the two PTA datasets separately and first neglect the possible presence of an astrophysical GW background from SMBHBs and employ the full Hellings-Downs (HD) ORF. We restrict this analysis to broad spectra and return later to narrow spectra. We choose logarithmic priors $\log_{10}\Delta \in [\log_{10}(0.5), \log_{10} 3], \log_{10} A_\zeta \in [-3, -1.22], \log_{10} k_\star/\text{Mpc}^{-1} \in [4, 9]$. The upper limit on the curvature power spectrum amplitude is dictated by the $f_{\text{PBH}} \leq 1$ constraint for $\Delta = 3$ and $k_\star = 10^5$ Mpc$^{-1}$ (for top-hat window function), that is the least constraining choice given the priors on $\Delta$ (constraints are stronger for smaller widths, see Fig. 6). One- and two-dimensional posterior distributions are reported in Fig. 3. Let us first focus on the right panel, where the posterior for the peak width is shown. We note that NG12 accommodates any value of $\Delta$, although it shows mild preference for broader peaks. This is expected, since for large $\Delta$ the GW spectrum is essentially flat in the NG12 range and the results of [2] are reproduced. On the other hand, IPTA DR2 more strongly prefers $\Delta \lesssim 1$, as well as $k_\star \gtrsim 3\cdot 10^6$ Mpc$^{-1}$. For such values of $k_\star$, the constraint from PBH overproduction is significantly stronger than the upper prior imposed in our search. We show the curve $f_{\text{PBH}} = 1$ for $\Delta = 1$ in the right panel of Fig. 3. While the strength of the constraint depends on the peak width $\Delta$, which the posteriors in the right panel of Fig. 3 have been marginalized over, one should notice that most of the $\Delta$ posterior for IPTA DR2 sits precisely in the $\Delta \leq 1$ region, where the constraint would be even stronger than what is shown in the figure. This analysis thus serves the purpose of showing that the scalar induced only GWs interpretation of the IPTA DR2 common-spectrum process is strongly affected by cosmological constraints. Indirect constraints from astrophysical bounds on $f_{\text{PBH}}$ are also shown (see App. B) by the dotted black curve, as well as constraints from LIGO/Virgo, obtained by translating the bounds of [79] on $f_{\text{PBH}}$. The resulting maximum likelihood GW spectra are shown in Fig. 4 as solid curves, together with the free spectrum posteriors obtained by appropriately converting the results of [2,8]. The corresponding integrated amplitude $A_\zeta$ is actually very similar for both NG12 and IPTA DR2, although the smaller peak

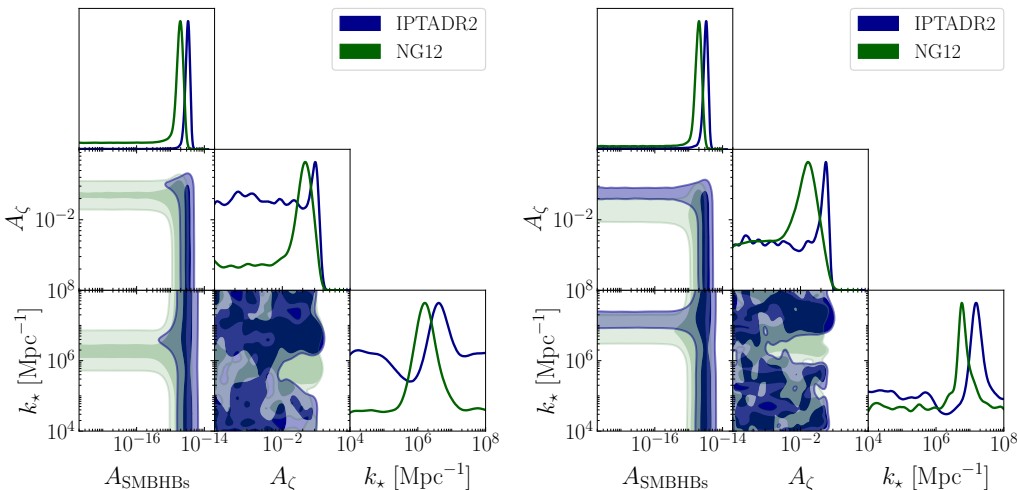

Figure 5: 1- and 2-d Posterior distributions for the stochastic gravitational wave background sourced by curvature perturbations and by SMBHBs. A conservative upper prior on $A_\zeta$ from overproduction of PBHs has been applied. The dark (light) shaded regions show 68% and 95% C.L. regions respectively. *Left*: $\Delta = 1$, with upper prior $\log_{10} A_\zeta \leq -1.44$. *Right*: $\Delta = 0.05$, with upper prior $\log_{10} A_\zeta \leq -1.57$.

width preferred by the latter causes a larger peak amplitude than for NG12. For IPTA DR2 (NG12), the excess is mostly fitted by the region at frequencies slightly smaller (larger) than the peak location.

We thus continue our detection analyses by including the expected stochastic gravitational wave background of astrophysical origin, from SMBHBs. Under the assumption of circular orbits and energy loss dominated by gravitational radiation, the characteristic strain of such GW background is expected to obey a simple power law: $h_c(f) = A_{\text{SMBHBs}}(f/\text{yr}^{-1})^{-2/3}$, see e.g. [11]. From now on, in order to reduce computational time, we consider only auto-correlation terms rather than the full Hellings-Downs (HD) ORF, following the NG12 [2] and IPTA DR2 [8] searches for common-spectrum processes. Since there is currently no evidence in favor nor against HD correlations in the datasets we consider, their inclusion does not significantly affect our results, especially when comparing GW models.

The total stochastic GW background is thus characterized by four parameters in total: $A_\zeta, k_\star$ (or alternatively $f_\star$) and $\Delta$ for the scalar induced spectrum, and $A_{\text{SMBHBs}}$ for the astrophysical background. Since the constraints from PBH overproduction vary significantly with the peak width $\Delta$, we choose to perform two analyses keeping the latter parameter fixed to two representative values $\Delta = 1, 0.05$ for the broad and narrow peak regime respectively. We fix the upper prior boundary for $A_\zeta$ to the value of the constraint for top-hat window function at $k_\star = 10^5$ Mpc$^{-1}$, i.e. $\log_{10} A_\zeta \leq -1.44(-1.57)$ for $\Delta = 1(0.05)$. As above, this choice corresponds to the weakest (most conservative) bound for the range of scales of interest (in other words, most scales in our search would actually be more constrained than what we are imposing). We impose logarithmic priors on the remaining parameters: $4 \leq \log_{10} k_\star/\text{Mpc}^{-1} \leq 9$, $-18 \leq \log_{10} A_{\text{SMBHBs}} \leq -13$.

The resulting posterior distributions are shown in Fig. 5. Let us focus on the NG12 dataset first (green-shaded regions). In the 2d distribution of the parameters $A_\zeta$ and $A_{\text{SMBHBs}}$, we observe two distinct regions for both $\Delta = 1$ and $\Delta = 0.05$, both allowed at 95%. The first region is centered around $A_{\text{SMBHBs}} \simeq 10^{-15}$ and covers all values of $A_\zeta$ up to the prior boundary. In this region the common-spectrum excess in the NG12 dataset is well-modeled by the GW background from SMBHBs only (indeed our 1d posterior for $A_{\text{SMBHBs}}$ is very similar to that

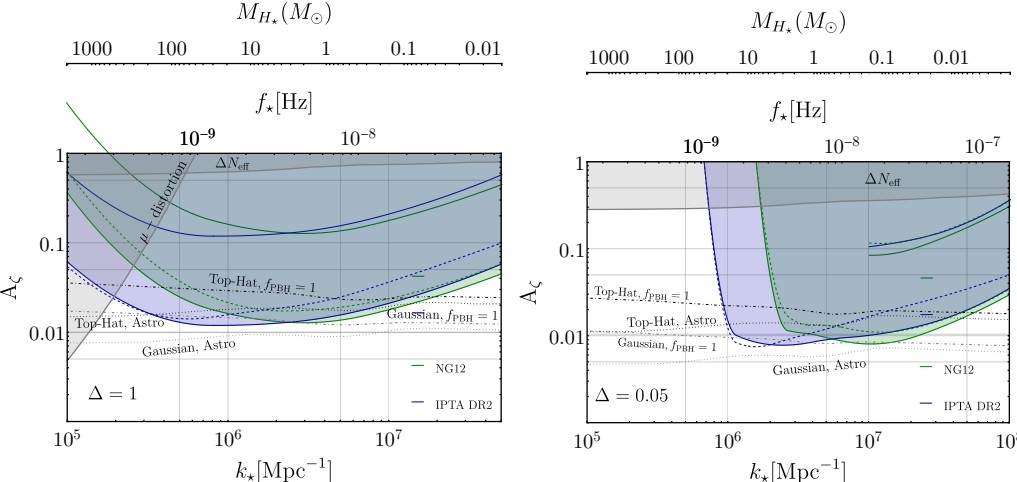

Figure 6: 95% C.L. upper limits on the amplitude of the curvature power spectrum (2), obtained by assuming the presence of an astrophysical GW background with $\log_{10} A_{\mathrm{SMBHBs}}$ fixed according to the posteriors of [2, 8]. *Left*: $\Delta = 1$. *Right*: $\Delta = 0.05$. The blue (green) shaded regions are constrained by IPTA DR2 (NG12), assuming the upper 95% C.L. posterior value $\log_{10} A_{\mathrm{SMBHBs}} = -14.4(-14.57)$. The dashed curves are obtained assuming the lower 95% C.L. posterior value $\log_{10} A_{\mathrm{SMBHBs}} = -14.7(-14.86)$ for IPTA DR2 (NG12) instead. Constraints from PBH overproduction are shown as black (gray) dot-dashed curves for a top-hat (Gaussian) window function, and have been obtained using $\delta_c = 0.46(0.59), \kappa = 4$ for the Top-Hat, and $\delta_c = 0.21(0.27), \kappa = 10$ for the modified Gaussian, for $\Delta = 1(0.05)$, see also App. B. Constraints from astrophysical observations are shown as black (gray) dotted curves for a top-hat (Gaussian) window function. The two gray-shaded regions are constrained by CMB observations: the upper region because GWs contribute to the effective number of neutrino species ($\Delta N_{\mathrm{eff}} \leq 0.28$ at 95% C.L. from Planck18+BAO [86]); the left corner region in the left panel because curvature perturbations cause $\mu$-distortions (constrained by COBE/FIRAS [47, 48], (see also [87] for a recent reassessment), see also App. B (the same constraints is shifted to smaller values of $k_\star$ in the right panel, out of the plot range). The frequency of GWs corresponding to $k_\star$ is shown in the first upper x-axis. The horizon mass at re-entry of the mode $k_\star$ is shown in the second upper x-axis.

of [2], only slightly broader due to the additional source of GWs in our search). The second region is instead centered around $A_\zeta \simeq 0.02 \,(0.015)$ for $\Delta = 1 \,(0.05)$ and spans all values of $A_{\mathrm{SMBHBs}}$ up to $A_{\mathrm{SMBHBs}} \lesssim 10^{-14}$. Here the excess is well-modeled by the scalar induced GWs only. Clearly, in the intersection of these two regions the excess is well-modeled by the combination of the two signals. Let us also examine the 2d distribution of the parameters $k_\star$ and $A_{\mathrm{SMBHBs}}$: the same pattern appears, with the scalar induced region now being centered around $k_\star \gtrsim 10^6 \,\mathrm{Mpc}^{-1}(\lesssim 10^7 \,\mathrm{Mpc}^{-1})$ for $\Delta = 1 \,(0.05)$. According to (6), such peak wavenumbers correspond to horizon masses $0.1 \, M_\odot \lesssim M_H \lesssim 20 \, M_\odot$ (the corresponding average PBH mass is larger by $O(1)$ factors, depending on the width of the spectrum, see App. B).

Let us now turn to the IPTA DR2 dataset (blue-shaded regions). As expected from the previous analysis, we notice a crucial difference with respect to the NG12 dataset: the region with $A_{\mathrm{SMBHBs}} \ll 10^{-15}$ is not allowed (at $2\sigma$ at least) in the broad peak case $\Delta = 1$. The reason for this is easy to understand: the amplitude of the common-spectrum process inferred in IPTA DR2 is larger than in NG12, therefore the amplitude of the scalar induced GW background should also be larger to provide a good modeling of the data. However, the upper prior from

PBH overproduction significantly limits this possibility. We have checked that this conclusion is independent of the value of $\Delta$ in the broad peak region, $\Delta \geq 0.5$, as long as the corresponding prior on $A_\zeta$ is used. A similar trend exists also in the narrow peak case $\Delta = 0.05$, although in this case the scalar induced regions are disfavored only at $1\sigma$.

In order to better assess whether there is any preference for one GW background over the other, we consider two models: one where the stochastic background is purely primordial and induced by scalar perturbations and another one where it is purely astrophysical.[4] We compare these models using the Bayes factors $\log_{10} B_{i,j}$ of model $j$ with respect to model $i$. For NG12, we find: $\log_{10} B_{\zeta,\text{SMBHBs}} \simeq 0.05(0.3)$ for $\Delta = 1(0.05)$. Therefore, we find no substantial evidence for one model against the other one in the NG12 dataset, as expected from the green shaded regions in Fig. 5. On the other hand, for IPTA DR2 we find $\log_{10} B_{\zeta,\text{SMBHBs}} \simeq 2.2$ for $\Delta = 1$, which implies decisive evidence for the SMBHBs model over the scalar induced model in the IPTA DR2 dataset. The evidence is weaker, though still substantial, for $\Delta = 0.05$: $\log_{10} B_{\zeta,\text{SMBHBs}} \simeq 0.9$. The maximum likelihood GW spectrum from SMBHBs obtained by this search is shown in Fig. 4 by the dashed curves.

Overall, our search reveals that the scalar induced GW interpretation is disfavored by IPTA DR2 data compared to the SMBHBs model, whereas NG12 data are fitted equally well by the two models. This conclusion is reached with a rather conservative prior choice and is thus robust; a more aggressive choice based on the $f_{\text{PBH}} \leq 1$ constraint applicable in the region $k_\star \gtrsim 10^6 \text{ Mpc}^{-1}$ is expected to constrain the scalar induced GW model in both datasets further (in fact the constraint at $k_\star = 10^6 \text{ Mpc}^{-1}$ reads $\log_{10} A_\zeta \leq 0.03(0.02)$ for $\Delta = 1(0.05)$, which very significantly constrains the IPTA DR2 region in the right panel of Fig. 5).

These results motivate a different type of analysis, aimed at setting upper limits to the amplitude $A_\zeta$ of curvature perturbations as a function of the peak scale $k_\star$. To this end, we proceed as follows: we fix the amplitude $A_{\text{SMBHBs}}$ of the astrophysical background to the value inferred by the SMBHBs analysis only of the NG12 and IPTA DR2 collaborations; we consider a set of values of $k_\star$ and obtain 95% C.L. upper limits on $A_\zeta$ for each peak location $k_\star$ in this set, keeping the width $\Delta$ fixed as above. Given that the collaborations report $2\sigma$ intervals: $\log_{10} A_{\text{SMBHBs}} \in [-14.86, -14.57]$ for NG12 [2] and $\log_{10} A_{\text{SMBHBs}} \in [-14.7, -14.4]$ for IPTA DR2 [8], we perform two analyses per dataset, for each value of $\Delta$ and $k_\star$, setting $A_{\text{SMBHBs}}$ to the interval boundaries of the corresponding PTA dataset (as given by the collaborations, notice also that these results are in good agreement with astrophyical expectations, see e.g. [8,12]). In this type of analysis, we do not impose an upper prior on $A_\zeta$ from PBH overproduction, as we are interested in independent constraints (see App. C for details on prior choices). An alternative strategy to set (weaker) upper limits consists in constraining the amplitude $A_\zeta$, without including any other GW signal in the analysis. In this case, the constraint would roughly follow the upper boundary of the $2\sigma$ region in Fig. 3, since any value of $A_\zeta$ above it leads to too strong GW signals. The stronger constraints derived in this paper are motivated by a theoretical and observational preference for a SMBHB contribution in the data once the contraints from PBH production are taken into account, which as we have shown importantly limit the possibility of scalar-induced GWs to model the common process in PTA data.

Results are reported in Fig. 6. As expected from Fig. 1, we observe stronger constraints for narrow peak spectra. However, the constrained range of $k_\star$ is wider in the broad peak case; this is partially caused by our restriction to the IR tail of the signal in the narrow peak case, but would be the case even including the full spectrum, since it decreases exponentially at frequencies only slightly larger than $f_\star$. For this reason, PTAs cannot provide any constraint

---

[4]For the primordial model, we impose an upper prior on $A_\zeta$ as for the previous search. However, in this search we are only interested in values of $k_\star$ for which the scalar induced GW background can fit the data in the absence of an astrophysical background. Therefore, the relevant range of wavenumbers is narrower than in the previous search, and starts at $k_\star \gtrsim 5 \cdot 10^5 \text{ Mpc}^{-1}$. We thus impose a tighter upper prior on $A_\zeta$, corresponding to the value of the $f_{\text{PBH}} = 1$ curve at $k_\star = 5 \cdot 10^5 \text{ Mpc}^{-1}$. A full recap of our choice of priors is presented in App. C.

on narrow peaked spectra for wavenumbers close to and smaller than those corresponding to the first bins of the datasets (notice the sharp cut of the constraint regions for $\Delta = 0.05$).

For IPTA DR2, the strongest constraints in the broad peak case are obtained for wavenumbers corresponding to the peak sensitivity of the PTA, see [8]. NG12 provides the stronger constraints at larger wavenumbers. This is expected, since the first bin of the NG12 dataset sits at $f \simeq 2.5 \cdot 10^{-9}$ Hz, whereas IPTA's first bin is at $f \simeq 10^{-9}$ Hz. In the regions where the constraints overlap, they are of comparable magnitude.

The difference between solid and dashed curves can be taken as an uncertainty on the constraints, given that it corresponds to the uncertainty on the common-spectrum process parameter $A_{\text{SMBHBs}}$. We notice that the dashed constraint intersects the solid curve at small frequencies for the IPTA DR2 dataset. This apparently contradictory feature may be caused by the fact that by lowering the amplitude of the astrophysical background, a component of the common-spectrum process may be explained by scalar induced GWs; however IPTA DR2 constrains the high-frequency tail (relevant for $f_\star \ll 10^{-9}$ Hz) of the GW spectrum sourced by scalar perturbations more strongly than the peak region (see [8] for power law posteriors). In other words, while in most of the parameter space a larger value for $A_{\text{SMBHBs}}$ leaves less room for a stochastic background from scalar perturbations, leading to stronger constraints, the situation is inversed for small values of $k_\star$ where the spectral shape (3) provides a poor fit to the data.

Constraints from PBH overproduction are also shown in Fig. 6, as dashed lines. We see that our constraints are significantly stronger than the overproduction limits obtained with the top-hat window function, whereas they are comparable to those obtained with a modified Gaussian window function.

We also report other constraints on $A_\zeta$, derived from astrophysical bounds on $f_{\text{PBH}}$ (see App. B for details), as dotted curves. These are obviously stronger than the overproduction constraints. In the broad peak case, at scales $5 \cdot 10^5$ Mpc$^{-1} \lesssim k_\star \lesssim 2 \cdot 10^7$ Mpc$^{-1}$ our strongest constraints can be stronger or slightly weaker than astrophysical bounds, again depending on the choice of window function. In the narrow peak case, this range is shifted to $10^6$ Mpc$^{-1} \lesssim k_\star \lesssim 5 \cdot 10^7$ Mpc$^{-1}$. Constraints from the scalar induced GW contribution to the effective number of neutrino species [86] (see also [80]) as well as from $\mu$-distortions [47,48] are also shown as shaded gray regions.

The horizon mass when at re-entry of the mode $k_\star$ is also shown in Fig. 6, see the uppermost x-axis. As mentioned above, the average PBH mass is only slightly larger than the horizon mass, therefore the scales constrained by PTAs correspond to PBHs with average masses $0.05\ M_\odot \lesssim M_{\text{PBH}} \lesssim 10^3\ M_\odot$ for broad spectra and $0.01\ M_\odot \lesssim M_{\text{PBH}} \lesssim 20\ M_\odot$ for narrow spectra. However, we stress once again that no reliable constraint on $f_{\text{PBH}}$ can be currently extracted by means of PTAs, given theoretical uncertainties related to the choice of window function.

Finally, two comments are in order. First, as mentioned in Sec. 2, we have limited our search to the low-frequency tail (starting roughly at the location of the dip in Fig. 1) of the GW spectrum for $\Delta < 0.5$, due to the resolution of PTAs. We have checked for $\Delta = 0.05$ that the results from the NG12 search using a smoothing strategy for the peak region are similar to those presented here, although slightly larger values of $k_\star$s are then allowed.[5] Our choice in this work is overall expected to slightly underestimate the total GW signal, therefore the constraints presented in Fig. 6 (right panel) are conservative. Second, we expect our constraints to remain valid even if the common-spectrum process observed at PTAs is not due to GWs, given that our analysis has been performed without including Hellings-Downs correlations.[6]

---

[5]In practice, we replaced the peak region by a plateau of amplitude set to the mean of $\Omega_{\text{gw}}h^2$ over that range of $f_\star$.
[6]A common red spectrum with slope $-2/3$ provides a good fit to both IPTA DR2 and NG12 data independently of its possible astrophysical origin.

# 5 Relation to previous works

The search for scalar induced GWs in PTA datasets has received increased attention over the past few years, with significant progress but also some apparently contradictory statements arising. In this section we clarify the relation of our findings with recent previous literature.

First, we comment on constraints on the curvature power spectrum from previous PTA datasets. Ref. [35] performs a search in the NANOgrav 11 years dataset. Differently from our choice, this work assumes that the power spectrum is given by a delta function, $\sim A k_\star \delta(k-k_\star)$, corresponding to $\Delta \to 0$ in (2). The resulting constraints on $A$ are comparable to our results for $\Delta = 0.05$ (we did not explore smaller peak widths, for which we expect slightly stronger constraints than for $\Delta = 0.05$, since as $\Delta \to 0$ the IR tail of the signal behaves as $f^2$ rather than $f^3$ across a larger frequency range). On the other hand, Ref. [35] also claims very strong constraints on $f_{PBH}$ which are reported in several other works (see e.g. [88]). As stressed above, such constraints suffer from the exponential sensitivity to the choice of window function and the use of the appropriate threshold. In particular, the very strong constraints presented in [35] rely on their choice for the critical threshold, $\delta_c = 1$, whereas explicit calculations point to a smaller value, see App. B. We checked that using values of the threshold close to the ones considered in our work and including the non-linear relation between $\delta_l$ and $\delta_m$, which was neglected in [35], very significantly weakens the constraints of from NG11 on $f_{PBH}$ (in particular it renders them weaker than current astrophysical constraints, which is consistent with our findings.)

Refs. [39,89] translate older PTA constraints (from 2015) on the stochastic GW background to constraints on the amplitude of a power-law $\sim (k/k_\star)^4$ or Gaussian curvature power spectrum respectively, while [34] uses the same strategy for a log-normal spectrum. The results of [34] can be directly compared to ours and they are of similar strength (after taking into account the different normalization). This apparently surprising feature is likely caused by the fact that limits on the stochastic GW background from older datasets are in tension with the current detection of a common-process spectrum in the latest datasets, signaling that they were likely too aggressive (see [2] for a discussion).

While our work is the first one to perform a bayesian search for scalar induced GWs in the IPTA DR2 dataset and, additionally, to account for the astrophysical background from SMBHBs, two papers have recently studied the implications of NG12 for scalar induced GWs. Rather than a bayesian search in the dataset, [45] uses the five bins free spectrum posteriors of [2] to find posteriors on the parameters of a broken power-law spectrum, similar to our log-normal spectrum for $\Delta \gtrsim 1$. Their values for the amplitude of the power spectrum are similar to ours (accounting for different normalizations) for NG12, as expected in the regime where the data is fit by a scalar induced SGWB which can be approximately modeled by a (broken) power law in the PTA range. On the other hand, their astrophysical constraints on $A_\zeta$ differ, most evidently because their bounds for a Gaussian window function are weaker than for a top-hat. We suspect that this is due to the choice of threshold in the Gaussian case (it seems that the threshold for a modified Gaussian is used, whereas a standard Gaussian is used as window function). As discussed above, the results for $A_\zeta$ are exponentially sensitive to this threshold.

On the other hand, [41] performs a bayesian search in the NG12 dataset, using a log-normal spectrum as we do. However, differently from our work, [41] includes all thirty frequency bins in the search. As pointed out in [2], this is problematic and leads to very different posteriors on common-spectrum process parameters compared to the five bins analysis adopted in our work following the NG12 search for a stochastic background [2]. We moreover disagree on the critical threshold used to recast the posteriors for the spectrum parameters to posteriors for the PBH fraction (too high for a Gaussian window function), see App. B.

Finally, three papers appeared shortly after the NG12 release, claiming that the NG12 common-spectrum excess could be explained by scalar induced GWs [42–44]. First, [43] assumes a log-normal power spectrum with $\Delta = 1$ and finds that solar mass PBHs may explain the NG12 excess, if the curvature power spectrum has amplitude $A_\zeta \sim 0.02-0.04$. Our NG12 posteriors in Fig. 5 agree with this conclusion, and actually allow for an even wider range of PBH masses. However, as argued above, the scenario is significantly disfavored compared to the astrophysical explanation in the IPTA DR2 dataset. Additionally, the computation of the PBH fraction may underestimate the PBH production, since a large threshold is used for a modified Gaussian window function (see appendix B). On the other hand, the non-linear relation between $\delta_m$ and $\delta_l$ is neglected.

Second, [42] finds that SMBHs with $M_\odot > 10^3 \, M_\odot$ may also explain the NG12 excess. These masses correspond to $k_\star \lesssim 10^5 \, \text{Mpc}^{-1}$ (see Fig. 2), which is disfavored at more than $2\sigma$ for a log-normal power spectrum by our analysis, using the NG12 dataset, and more significantly by IPTA DR2. However, [42] assumes a broken power-law curvature power spectrum, which induces a linearly decreasing GW spectrum at $f > f_\star$. In this case, one can simply use the power law results of [2,8]. Using a value of the critical threshold which is well-motivated for the broken power-law spectrum of [42] ($\delta_c \simeq 0.4-0.5$), we find that the supermassive PBHs interpretation ($M > 10^3 \, M_\odot$) of [42] is at best marginally allowed by cosmological constraints as an interpretation of the NG12 excess. It is however strongly disfavored by IPTA DR2 (and similarly by EPTA and PPTA), see the power-law posteriors in [8].

Third, [44] (see also [90]) considers a flat curvature power spectrum that extends from $k_l \simeq 10^5 \, \text{Mpc}^{-1}$ to $k_s \simeq 10^{15} \, \text{Mpc}^{-1}$, in such a way as to induce a broad PBH mass distribution peaked at masses for which PBHs can make all of the DM, $\simeq (10^{-16}-10^{-11}) \, M_\odot$. Results on this scenario can then be obtained by simply using the posteriors for power-law common-spectrum process presented in [2] and [8]. We notice in this respect that a flat spectrum is actually in $\simeq 2\sigma$ tension with the IPTA DR2 posteriors. More importantly, the amplitude inferred from IPTA DR2 is larger than from NG12. We then find that the IPTA DR2 lower bound on the amplitude of the power spectrum (even at $3\sigma$) is not compatible with the overproduction constraint $f_{\text{PBH}} \leq 1$, for any value of the cutoff scale $k_s$ in the PBH DM window. While modifying the proposal of [44] to a slightly red-tilted (rather than flat) curvature power spectrum is sufficient to make it viable with cosmological constraints, it does not alter the conclusions that such an almost flat slope is disfavored (at $\gtrsim 2\sigma$) by IPTA DR2 (and similarly by EPTA).

Finally, [91] considers the relation between NG12 and scalar induced GWs produced during a non-standard cosmological epoch dominated by a non-adiabatic fluid. Our results do not apply to this scenario, since the emission and propagation of GWs is affected by the background expansion of the Universe.

## 6  Conclusions

We presented searches for a scalar induced stochastic GW background in two of the most recent PTA datasets, focusing on the possibility of an an enhanced (with respect to CMB scales) curvature power spectrum at small scales $k \sim (10^5-10^8) \, \text{Mpc}^{-1}$. This is an especially interesting possibility, since it may also lead to the formation of PBHs with masses $\sim (0.001-1000) \, M_\odot$.

Since current data show strong evidence for a common-spectrum process, we have first focused on assessing the extent to which the excess can be modeled by scalar induced GWs, as proposed by several recent works after the NG12 release. To this aim, we have included three important novelties with respect to previous work. First, we have performed a search on the IPTA DR2 data set, in addition to a search on the NG12 data set. The former is known to favor a larger amplitude for the process than NG12, as well as a slightly positive (rather

than negative) slope for the spectrum. Second, we have taken into account constraints on the amplitude of the curvature power spectrum from the overproduction of PBHs, as priors in our searches. We have assessed them using a consistent choice of window function in the calculation of the variance and critical threshold for gravitational collapse and including the effects of the non-linear relation between matter and curvature perturbations. Thirdly, we have included the unavoidable stochastic GW background of astrophysical origin, from SMBHBs.

Our first main conclusions are: 1) the overproduction of PBHs associated with the large curvature perturbations significantly constrains the scalar induced interpretation of the IPTA DR2 common-spectrum process; 2) the astrophysical origin is favored over the scalar induced primordial origin by the IPTA DR2 dataset. This conclusion is stronger for broad power spectra, but remains valid for narrow spectra as well. On the other hand, we found that the NG12 dataset does not prefer any model over the other one. This difference in the results reflects the mild disagreement between the datasets ($\gtrsim 2\sigma$) reported by IPTA DR2 for a power-law common-spectrum process [8] (we notice that EPTA and PPTA latest releases agree well with IPTA DR2 on the slope of the spectrum). We have discussed the impact on previous proposals to interpret the common-spectrum process in PTA datasets in terms of scalar induced GWs, such as [42–44]. We reached our conclusions by using conservative (i.e. arguably weaker than their realistic value) prior choices on the amplitude of the power spectrum from PBH overproduction.

Motivated by our findings, we set constraints on the amplitude of the curvature power spectrum at scales $k \sim 10^5 - 10^8$ Mpc$^{-1}$. These are the most up-to-date constraints from PTAs, and are importantly independent from indirect astrophysical bounds on PBHs of masses $(0.05 - 1000)$ $M_\odot$ (dotted lines in Fig. 6), which suffer from theoretical uncertainties on the calculation of the PBH relic abundance. Our constraints are nonetheless already competitive with those bounds (a precise comparison depends on the choice of window function to obtain the astrophysical bounds). They are also significantly stronger (roughly by a factor of six) than similar constraints from LIGO/Virgo/KAGRA at much smaller scales ($k \gtrsim 10^{15}$ Mpc$^{-1}$) [46].

Our work also clarifies some inconsistencies in previously derived constraints on PBHs from PTAs, which we find to be largely due to the exponential sensitivity of the PBH relic abundance on the choice of the window function and on the threshold for PBH formation. Regarding the former, we estimate the uncertainty by providing results for different choices of the window function, for the latter we carefully ensure a consistent choice of window function and threshold value across our analysis.

In the next years, upcoming PTA results from NG, PPTA and EPTA will shed light on the origin of the common-process spectrum in current datasets. If evidence of Hellings-Downs correlation arises, it will be crucial and exciting to understand the origin of the signal, which can be sourced by several well-motivated phenomena in the early Universe, in addition to the astrophysical background from SMBHBs. Our work shows that one such mechanism can be effectively probed and constrained by PTAs (independently of whether the currently detected process is indeed due to GWs), and highlights the importance of complementary constraints from cosmology. It also provides an important step for future PTA data releases, that are expected to provide the strongest constraints on the curvature power spectrum at the epoch of the QCD crossover.

# Acknowledgments

We thank Sebastian Clesse and Nicholas Rodd for useful discussions. We also thank G. Franciolini, I. Musco, A. Urbano and P. Pani for correspondence on the use of peak theory, and H. Veermae and V. Vaskonen for comments on a first version of this paper.

**Funding information** The work of F.R. is partly supported by the grant RYC2021-031105-I from the Ministerio de Ciencia e Innovación (Spain). This project has partially received support from the European Union's Horizon 2020 research and innovation programme under the Marie Sklodowska-Curie grant agreement No 860881- HIDDeN. V. Dandoy thanks CERN for hosting a research stay during which this project was initiated.

# A Gravitational wave spectrum

The aim of this appendix is to provide the expressions for the scalar induced GW signal which we used in our work. In the radiation domination era, the GW spectrum averaged over many oscillation periods $\Omega_{\text{gw,r}}(k)$[7] is given as a function of the curvature power spectrum by [28, 49, 51],

$$\Omega_{\text{gw,r}}(k) = 3 \int_0^\infty d\,v \int_{|1-v|}^{1+v} d\,u \frac{\mathcal{T}(u,v)}{u^2 v^2} P_\zeta(uk) P_\zeta(vk), \tag{A.1}$$

with the transfer function $\mathcal{T}$ given by,

$$\mathcal{T}(u,v) = \frac{1}{4} \left[ \frac{4v^2 - (1+v^2-u^2)^2}{4uv} \right]^2 \left( \frac{u^2+v^2-3}{2uv} \right)^4 \\ \times \left[ \left( \log \frac{|3-(u+v)^2|}{|3-(u-v)^2|} - \frac{4uv}{u^2+v^2-3} \right)^2 + \pi^2 \Theta\left(u+v-\sqrt{3}\right) \right]. \tag{A.2}$$

After matter-radiation equality, the GW energy density decays as radiation such that the current value of the spectrum is

$$\Omega_{\text{gw}} h^2(f) \simeq 10^{-5} \left( \frac{g_*(T_\star)}{17.25} \right) \left( \frac{g_{*s}(T_\star)}{17.25} \right)^{-\frac{4}{3}} \left( \frac{\Omega_{r,0} h^2}{4 \times 10^{-5}} \right) \Omega_{\text{gw,r}}(f). \tag{A.3}$$

For a general shape of the curvature spectrum this can only be evaluated numerically. Nevertheless, the following two approximations of (A.1) were derived for the narrow ($\Delta \ll 0$) and broad ($\Delta \gg 0$) peak regimes of the log-normal curvature spectrum defined in (2) in [49]:

*Narrow peak*

$$\frac{\Omega_{\text{gw,r}}(f/f_\star, \Delta)}{A_\zeta^2} \approx$$

$$3\alpha^2 e^{\Delta^2} \left[ \text{erf}\left( \frac{1}{\Delta} \text{arcsinh} \frac{\alpha e^{\Delta^2}}{2} \right) - \text{erf}\left( \frac{1}{\Delta} \text{arcosh} \frac{\alpha e^{\Delta^2}}{2} \right) \right] \left( 1 - \frac{1}{4}\alpha^2 e^{2\Delta^2} \right)^2 \left( 1 - \frac{3}{2}\alpha^2 e^{2\Delta^2} \right)^2$$

$$\times \left\{ \left[ \frac{1}{2} \left( 1 - \frac{3}{2}\alpha^2 e^{2\Delta^2} \right)^2 \log\left| 1 - \frac{4}{3\alpha^2 e^{2\Delta^2}} \right| - 1 \right]^2 + \frac{\pi^2}{4} \left( 1 - \frac{3}{2}\alpha^2 e^{2\Delta^2} \right)^2 \Theta\left( 2 - \sqrt{3}\alpha e^{\Delta^2} \right) \right\}. \tag{A.4}$$

*Broad peak*

---

[7]The index $, r$ follows the notation of Ref. [49] and highlights that the quantity $\Omega_{\text{gw,r}}(k)$ is the time independent spectrum.

$$\frac{\Omega_{\text{gw,r}}(f/f_\star, \Delta)}{A_\zeta^2} \approx$$

$$\frac{4}{5\sqrt{\pi}} \alpha^3 \frac{e^{\frac{9\Delta^2}{4}}}{\Delta} \left[ \left( \log^2 K + \frac{\Delta^2}{2} \right) \text{erfc}\left( \frac{\log K + \frac{1}{2}\log\frac{3}{2}}{\Delta} \right) - \frac{\Delta}{\sqrt{\pi}} \exp\left( -\frac{\left(\log K + \frac{1}{2}\log\frac{3}{2}\right)^2}{\Delta^2} \right) \right.$$

$$\left. \times \left( \log K - \frac{1}{2}\log\frac{3}{2} \right) \right] + \frac{0.0659}{\Delta^2} \alpha^2 e^{\Delta^2} \exp\left( -\frac{\left(\log\alpha + \Delta^2 - \frac{1}{2}\log\frac{4}{2}\right)^2}{\Delta^2} \right)$$

$$+ \frac{1}{3}\sqrt{\frac{2}{\pi}} \alpha^{-4} \frac{e^{8\Delta^2}}{\Delta} \exp\left( -\frac{\log^2\alpha}{2\Delta^2} \right) \text{erfc}\left( \frac{4\Delta^2 - \log\alpha/4}{\sqrt{2}\Delta} \right),$$

$$(A.5)$$

where $\alpha = f/f_\star$ and $K = \alpha \exp\left(3\Delta^2/2\right)$.

As discussed in the main text, for the narrow peak case, this approximation deviates from the full numerical expression in the IR region by a constant factor (see Fig. 1). For this reason we modified the narrow peak approximation of [49] by correcting (A.4) as

$$\Omega_{\text{gw,r}}(f/f_\star, \Delta) \to \Omega_{\text{gw,r}}(f/f_\star, \Delta) \times \frac{1}{4} \left\{ 2 + \left(1 + \tanh\left[\left(-3\Delta e^{-\Delta^2} + k_\star\right)/(2\Delta)\right]\right) \right\}. \quad (A.6)$$

This correction corresponds to smoothly turning on the missing constant factor using a tanh function located at the transition scale between the $f^2$ and $f^3$ behavior of the gravitational wave spectrum.

# B Derivation of constraints

Here we review the basics of PBH formation from the collapse of curvature perturbations, and provide details on our determination of the PBH overproduction constraint.

## B.1 Primordial black hole formation

Let's consider a spherically symmetric density contrast $\delta(r) = \delta\rho/\rho_b$ initially at superhorizon scales characterized by a physical radius $R_m$.[8] We define its volume averaged perturbation $\delta_m$ via a smoothing function as

$$\delta_m = \int_0^\infty dR \, 4\pi R^2 \, \frac{\delta\rho}{\rho_b}(R, t_H) \, W(R; R_m), \quad (B.1)$$

where $t_H$ is the horizon crossing time and $W(R, R_m)$ is the window function used to smooth over the perturbation scale. If at horizon crossing the volume averaged perturbation $\delta_m$ exceeds the threshold $\delta_c$ (defined below), gravity forces overcome pressure forces and the perturbation collapses into a black hole [58, 63, 66]. Its mass is then found to be [61, 62, 92]

$$M(\delta_m) = \kappa M_H(r_m)(\delta_m - \delta_c)^\gamma, \quad (B.2)$$

where $M_H(r_m)$ is the horizon mass at a scale $k = 1/r_m$ and $\gamma$ and $\kappa$ are constant parameters (see main text for explicit expressions).

---

[8]In the rest of the calculation, the physical radial coordinate will be written with capital $R$ and the comoving one with small $r$.

In order to collapse, the amplitude of the density contrast must be relatively large, making it necessary to express $\delta_m$ in terms of the curvature perturbation $\zeta$ beyond linear order [58–60]. A non linear expression of $\delta_m$ is obtained by using the non-linear relation between the density contrast $\delta\rho/\rho_b$ and the curvature perturbation $\zeta$ (see Refs. [66,93,94])

$$\frac{\delta\rho}{\rho_b}(r,t) = -\frac{4}{3}\Phi(t)\left(\frac{1}{aH}\right)^2 e^{-5\zeta(r)/2}\nabla^2 e^{\zeta(r)/2}, \tag{B.3}$$

where $\Phi(t)$ depends on the equation of states of the Universe (see Ref. [71]) and is given by $\Phi = 2/3$ in a radiation fluid.

For a top-hat window function and using the last expression, it is easy to see that at linear order, the volume averaged density $\delta_m$ is related to the curvature perturbation by

$$\delta_m = -2\Phi r_m \zeta'(r_m) \equiv \delta_l, \tag{B.4}$$

where we defined $\delta_l$ as the smooth density contrast at linear order. With the full non-linear relation, we get

$$\delta_m = \left(\delta_l - \frac{1}{4\Phi}\delta_l^2\right). \tag{B.5}$$

With this it is then possible to rewrite the black hole mass as a function of the linear density contrast,

$$M(\delta_l) = \kappa M_H(r_m)\left(\delta_l - \frac{1}{4\Phi}\delta_l^2 - \delta_c\right)^{\gamma}. \tag{B.6}$$

Let's finally note that the constants $(\kappa, \delta_c)$ will be modified in this last expression if a modified Gaussian window function $W \sim \exp(-(R_m/R)^2/4)$ is initially used in Eq. (B.1). Detailed calculations [63,95] for a wide range of curvature power spectra showed that the critical threshold $\delta_c$ and $\kappa$ are modified as follows:

$$\begin{aligned}(\delta_c)^{\mathrm{TH}} &\approx 2.17 \times (\delta_c)^{\mathrm{G}}, \\ (\kappa)^{\mathrm{TH}} &\approx \frac{4}{2.74^2 \times 2.17^{\gamma}}(\kappa)^{\mathrm{G}},\end{aligned} \tag{B.7}$$

where "TH" stays for Top-Hat and "G" for modified Gaussian.

## B.2 PBH distribution from Press Schechter formalism

The Press Schechter formalism [55] is usually used to calculate the PBH population produced by a given curvature power spectrum $P_\zeta$. It is typically assumed that the probability distribution for the linear density perturbation $\delta_l$ at a scale $k$ is Gaussian and given by

$$P_k(\delta_l) = \frac{1}{\sqrt{2\pi\sigma_k^2}}\exp\left(-\frac{\delta_l^2}{2\sigma_k^2}\right). \tag{B.8}$$

The variance $\sigma_k^2$ at this scale is determined by the window function (which should coincide with the choice in (B.1)) and the curvature power spectrum [42,56,58,63]

$$\begin{aligned}\sigma_k^2 &= \langle\delta_l^2\rangle \\ &= \int_0^\infty \frac{dk'}{k'} W^2(k';k)P_\delta(k') \\ &= \frac{4}{9}\Phi^2 \int_0^\infty \frac{dk'}{k'}(k'/k)^4 T^2(k',k)W^2(k';k)P_\zeta(k'),\end{aligned} \tag{B.9}$$

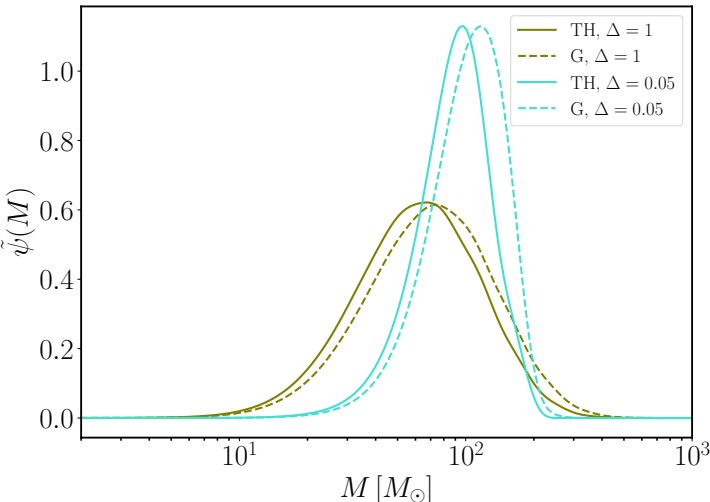

Figure 7: Normalized mass function $\tilde{\psi}(M) = \psi(M)/\Omega_{\text{PBH}}$ for a log-normal curvature power spectrum peaked at $k_\star = 10^6$ Mpc$^{-1}$ ($M_H \approx 20M_\odot$) and an amplitude $A_\zeta$ chosen to return $f_{\text{PBH}} = 1$ (note however that the amplitude has negligible impact on the normalized mass function). As observed in Ref. [56], the mean mass increases as the width of the spectrum decreases (the turquoise and olive lines are respectively for $\Delta = 0.05$ and $\Delta = 1$). For consistent choices of thresholds, the choice of the window function has little impact on the mass distribution (the solid and dashed lines are respectively for the top-hat (TH) and Gaussian (G) window functions).

where $P_\delta$ is the density contrast power spectrum, $P_\zeta$ the curvature power spectrum and $T(k', k)$ the transfer function taking into account the damping of the modes at sub-horizon scales. The fraction of the total energy density $\beta_k(M)$ collapsing into black holes of mass $M$ when the scale $k^{-1}$ crosses the horizon is given by [42, 55, 56, 96]

$$\beta_k(M) = \int_{\delta_c}^{\infty} d\delta_l \frac{M(\delta_l)}{M_H(k)} P_k(\delta_l) \delta_D \left[ \ln \frac{M}{M(\delta_l)} \right], \tag{B.10}$$

where $M(\delta_l)$ is given by Eq. (B.6), $\delta_D$ is the Dirac delta function, $\delta_l(M) = 2\Phi \left( 1 - \sqrt{1 - \frac{1}{\Phi} \left( \delta_c + q^{1/\gamma} \right)} \right)$ and $q = M/(\kappa M_H(k))$.

The present day PBH mass distribution is then given by (see e.g. [42])

$$\psi(M) = \int d\log(k) \beta_k(M) \frac{\rho_\gamma(T_k)}{\rho_c^0} \frac{s^0}{s(T_k)}. \tag{B.11}$$

The function $\psi(M)$ is normalized such that $\int d\log M \, \psi(M) = \Omega_{\text{PBH}}$. An example of mass distribution is shown in Fig. 7 for a log-normal power spectrum peaked at $k_\star = 10^6$ Mpc$^{-1}$ (corresponding to $M_{H_\star} \approx 20M_\odot$) and amplitudes $A_\zeta$ chosen such that the PBH production accounts for the whole dark matter abundance. Notice that the mean mass is slightly larger than the horizon mass at $k_\star$. This is the case because the variance $\sigma_k$ is peaked at slightly smaller scales than $k_\star$ [43, 56]. For instance, with $k_\star = 10^6$ Mpc$^{-1}$ the mean mass for $\Delta = 1(0.05)$ is $M \approx 60(100)M_\odot$. Those number are in agreement with the horizon mass ratio given in Ref. [56] for such widths.

As mentioned in the main text, the calculated abundance depends strongly on the choice of the window function as well as on the choice of the threshold value $\delta_c$ [42, 56, 63, 68]. Detailed studies of the dependence of $\delta_c$ on the curvature power spectrum shape have been conducted

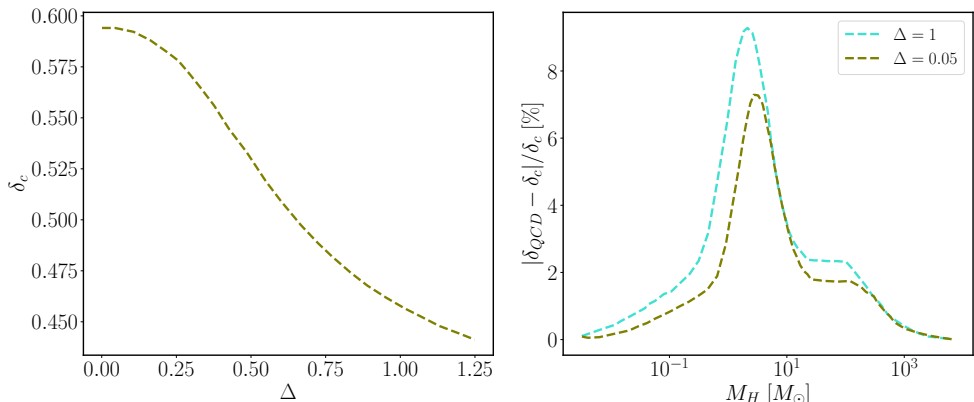

Figure 8: *Left*: Variation of the threshold $\delta_c$ as a function of the curvature power spectrum width $\Delta$ for a top-hat window function (see Ref. [64]). *Right*: Impact on the threshold from QCD effect as a function of the horizon mass $M_H$ (see Ref. [70]).

in [64–67] and found $\delta_c = \delta_c(\Delta)$ with $\delta_c \approx 0.6 - 0.4$ for $\Delta \approx 0 - 1$ (see Fig. 8) for the log-normal power spectrum considered in our work. Note however that those values have been derived for a top-hat window function. Their values for a Gaussian window function can be found by means of (B.1). Finally, around the QCD phase transition, the equation of state deviates slightly from $w = 1/3$. This drop provides an enhancement of PBH formation manifested by a reduction of the threshold at those scales [69, 70] (see right panel of Fig. 8 for the variation of the threshold as a function of the horizon mass).

### B.3   Constraints on the curvature power spectrum

In the range of masses considered in this paper the most important constraints on $f_{\mathrm{PBH}}$ come from microlensing [97–101], PBH merger rates as deduced by LIGO-VIRGO collaboration (see [79][9] and from PBH accretion signatures in CMB (assuming spherical accretion [102, 103]). All the constraints we employ are reviewed in Ref. [104]. These are however often derived using a monochromatic mass function. We used instead the method developed in Ref. [105] to deal with extended mass functions. Namely, if the observational constraints are represented by the function $f_{\mathrm{PBH}}(M)$ for monochromatic PBH mass $M$, the translation into an extended mass spectrum is given by

$$\int \mathrm{d}\log M \, \frac{\psi(M; A_\zeta, k_\star, \Delta)}{\Omega_{\mathrm{DM}} f_{\mathrm{PBH}}(M)} \leq 1 \,, \tag{B.12}$$

where $\psi(M; A_\zeta, k_\star, \Delta)$ is the mass function calculated using the formalism of the previous section for the curvature power spectrum $P_\zeta(k; A_\zeta, k_\star, \Delta)$ defined in Eq. (2). In the main text, we fix the spectrum width (two different values are taken $\Delta = 1$ and $\Delta = 0.05$), and upper limits on $A_\zeta$ as function of $k_\star$ are obtained by solving numerically the equation above. Similarly, the absolute limit $f_{\mathrm{PBH}} = 1$ is translated into constraints on $A_\zeta(k_\star)$ by simply setting $f_{\mathrm{PBH}}(M) = 1$ in Eq. (B.12). Results are obtained setting $\kappa = 4$ and $\gamma = 0.36$ for the top-hat window function (see Eq. (B.1) for the corresponding value if a modified Gaussian window function is used instead), as motivated by simulations [58]. Larger values of $\kappa$ lead to stronger

---

[9]We used the constraint obtained assuming that all BHs observed by LIGO/Virgo are astrophysical. Allowing for a primordial fraction has a minor effect on our constraint.

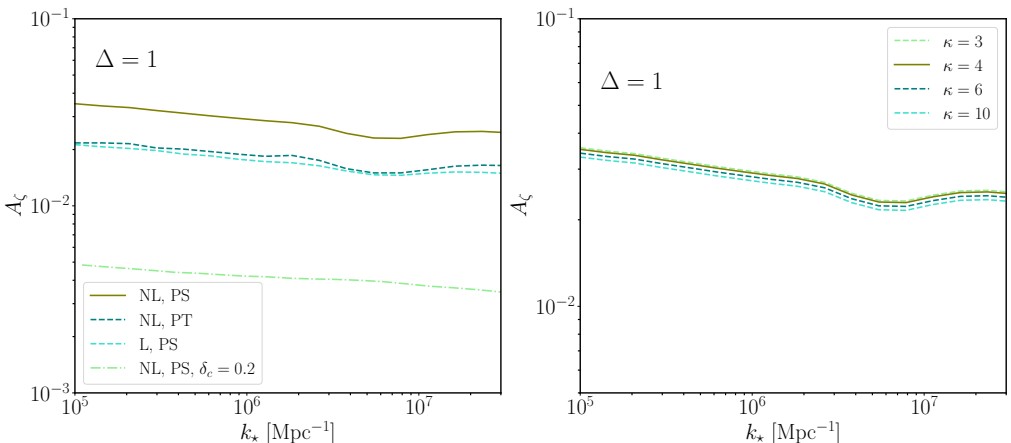

Figure 9: *Left*: Effect on the limit $f_{\mathrm{PBH}} = 1$ considering the Press-Schecter formalism (PS) with the non-linear (NL) and linear (L) relation for the density. Results using Peak Theory (PT) and with a lower threshold ($\delta_c = 0.2$) are also shown. *Right*: Similar but changing this time only the parameter $\kappa$. Both are calculated for a lognormal power spectrum with width $\Delta = 1$ and top-hat window function.

constraints (significantly smaller values do not seem to be supported by numerical studies). In Fig.9, we show how changing $\kappa$ or $\gamma$ has little impact on the $f_{\mathrm{PBH}} = 1$ constraint. As stressed in the main text, the constraints depend on the choice of the threshold as well as on the non-linear corrections. Fig. 9 shows how the limit for $f_{\mathrm{PBH}} = 1$ is altered by a change of critical threshold or neglecting the non-linearities. Finally, an uncertainty remains on the formalism used to calculate the PBH abundance. While in this work the Press-Schechter formalism has been used, it is known that using Peak Theory [106] instead increases the PBH production, thereby leading to stronger constraints [56]. We show how using Peak Theory modifies our $f_{\mathrm{PBH}} = 1$ limit in Fig.9.

Limits on the curvature power spectrum amplitude from scalar induced GWs can also be derived from CMB measurements. Gravitational waves behave as additional relativistic degrees of freedom beyond neutrinos, and are thus constrained by current bounds on the effective number of neutrino species $\Delta N_{\mathrm{eff}} \equiv N_{\mathrm{eff}} - 3.046$, where $N_{\mathrm{eff}} \equiv \rho_{\mathrm{gw}}/\rho_{\nu}$ in our case, and $\rho_{\nu}$ is the energy density of one neutrino species (for the CMB, quantities are evaluate at the epoch of recombination) [86]. In particular, the combination of Planck2018 + BAO gives $\Delta N_{\mathrm{eff}} < 0.28$ [86], which translates into constraints on the total amount of gravitational waves

$$\int d\log(f)\, h^2 \Omega_{\mathrm{gw}}(f; A_\zeta, k_\star, \Delta) < 5.6 \times 10^{-6} \Delta N_{\mathrm{eff}}, \tag{B.13}$$

where $\Omega_{\mathrm{gw}}(f; A_\zeta, k_\star, \Delta)$ is the gravitational wave spectrum produced from the curvature power spectrum defined in (2).

Finally, CMB measurements also set strong constraints on the amplitude of the curvature power spectrum at scales $k \lesssim 10^5\,\mathrm{Mpc}^{-1}$, since large perturbations cause $\mu$-distortions in the photon spectrum [107]. The commonly used parameter $\mu$ can be expressed in terms of the curvature power spectrum as

$$\mu \approx \int_{1\mathrm{Mpc}^{-1}}^{\infty} \frac{dk}{k} P_\zeta(k) W_\mu(k), \tag{B.14}$$

with the window function

$$W_\mu(k) \approx 2.27 \left[ \exp\left( -\frac{(k/1360)^2}{\left(1 + (k/260)^{0.3} + k/340\right)} \right) - \exp\left( -\left(\frac{k}{32}\right)^2 \right) \right]. \tag{B.15}$$

Observations from COBE/FIRAS [47,48] set the upper limit $\mu < 9 \times 10^{-5}$. Using our log-normal power spectrum with fixed $k_\star$ and width $\Delta$, we can then set constraints on the amplitude $A_\zeta$.

## C Numerical strategy

The aim of this Appendix is to provide details on our Bayesian search. We use the datasets released in [108] for NG12 and in [109] for IPTADR2 (Version B, we use par files with TDB units) and followed closely the strategy of the NG and IPTA collaborations for noise parameters, (we reproduced the results of [2,8] with excellent agreement).

Specifically, for both datasets we consider three types of white noise parameters per backend/receiver (per pulsar): EFAC ($E_k$), EQUAD ($Q_k[s]$) and ECORR ($J_k[s]$), the latter only for pulsars in the NG12 dataset and for NG 9 years pulsars in the IPTADR2 dataset. We also included two power-law red noise parameters per pulsar in both datasets: the amplitude at the reference frequency of $\mathrm{yr}^{-1}$, $A_{\mathrm{red}}$, and the spectral index $\gamma_{\mathrm{red}}$. For the IPTA DR2 dataset, we additionally included power-law dispersion measures (DM) errors (see e.g. [8]) (in the single pulsar analysis of PSR J1713+0747 we also included a DM exponential dip parameter following [8]).

We fixed white noise parameters according to their maximum likelihood a posteriori values from single pulsar analyses (without GW parameters). For the NG12 dataset (45 pulsars with more than 3 years of observation time), the white noise dictionary is provided in [108]. For IPTADR2, we used the dictionary built in [18] by performing single pulsar analyses for each pulsar with more than 3 years of observation time (for a total of 53 pulsars). The Jet Propulsion Laboratory solar-system ephemeris DE438 and the TT reference timescale BIPM18 have been used.

We perform two types of analyses in our work: First, as in [2,8], we perform detection analyses aimed at determining the region of parameter space for which scalar induced GWs can model the common-spectrum process in the datasets. Second, we also perform an upper-limit analysis to constrain the amplitude of the curvature power spectrum. The choice of priors for both noise (except for single pulsar white-noise parameters) and GW parameters is slightly different for the two strategies, as described in [108] (in upper-limit analyses a "Linear-Exponent" prior of the form $p(x) \propto 10^x$ is used, rather than a uniform prior on the logarithm of e.g. the GW amplitude from SMBHBs). All prior choices are reported in Table 1 and Table 2 for our detection and upper-limit analyses respectively. The specific prior choices for $A_\zeta$ are due to constraints from PBH overproduction and are motivated in Sec. 4 of the main text.

As in [2,8], for most of our runs we use only auto-correlation terms in the Overlap Reduction Function (ORF) in our search, rather than the full Hellings-Downs (HD) ORF, to reduce the computational time. On the other hand, we include the full HD ORF in our search for scalar induced GWs only, see posteriors in Fig. 3 (the computation of the Bayes factor is instead based on the analysis without HD correlations).

We obtain $5 \cdot 10^6$ samples for our detection analyses and discard 25% of each chain as burn-in (for the HD analysis, we collect roughly $10^6$ samples). For the upper-limit analyses, we collect $10^6$ samples and discard 10% of each chain. We consider the following set of values of $k_\star$ for constraints from IPTA DR2: $k_\star = (10^5, 6 \cdot 10^5, 10^6, 5 \cdot 10^6, 10^7, 5 \cdot 10^7)$ Mpc$^{-1}$ for $\Delta = 1$ and $k_\star = (6.8 \cdot 10^5, 9.5 \cdot 10^5, 1.4 \cdot 10^6, 3 \cdot 10^6, 5 \cdot 10^6, 10^7, 5 \cdot 10^7, 10^8)$ Mpc$^{-1}$ for $\Delta = 0.05$. Similarly, for NG12 we take the same set as for IPTA DR2 for $\Delta = 1$, and $k_\star = (1.6 \cdot 10^6, 2.2 \cdot 10^6, 3.2 \cdot 10^6, 4 \cdot 10^6, 5 \cdot 10^6, 5 \cdot 10^6, 10^7, 5 \cdot 10^7, 10^8)$ Mpc$^{-1}$ for $\Delta = 0.05$. The continuous curves shown in Fig. 6 are then obtained as smooth interpolations.

Table 1: List of noise and astrophysical GW background parameters used in our detection analyses, together with their prior ranges.

| Parameter | Description | Prior | Comments |
|---|---|---|---|
| **Detection analysis** | | | |
| Parameter | Description | Prior | Comments |
| **White Noise** | | | |
| $E_k$ | EFAC per backend/receiver system | Uniform $[0,10]$ | single-pulsar only |
| $Q_k[s]$ | EQUAD per backend/receiver system | log-Uniform $[-8.5,-5]$ | single-pulsar only |
| $J_k[s]$ | ECORR per backend/receiver system | log-Uniform $[-8.5,-5]$ | single-pulsar only (NG12, NG9) |
| **Red Noise** | | | |
| $A_{\rm red}$ | Red noise power-law amplitude | log-Uniform $[-20,-11]$ | one parameter per pulsar |
| $\gamma_{\rm red}$ | Red noise power-law spectral index | Uniform $[0,7]$ | one parameter per pulsar |
| **DM Variations Gaussian Process Noise** | | | |
| $A_{\rm DM}$ | DM noise power-law amplitude | log-Uniform $[-20,-11]$ | one parameter per pulsar (IPTADR2) |
| $\gamma_{\rm DM}$ | DM noise power-law spectral index | Uniform $[0,7]$ | one parameter per pulsar (IPTADR2) |
| **scalar induced GW Background, w/ SMBHBs** | | | |
| $A_\zeta, \Delta = 1$ | Power spectrum amplitude | log-Uniform $[-3,-1.44]$ | one parameter for PTA |
| $A_\zeta, \Delta = 0.05$ | Power spectrum amplitude | log-Uniform $[-3,-1.57]$ | one parameter for PTA |
| $k_\star[{\rm Mpc}^{-1}]$ | Peak scale of the power spectrum | log-Uniform $[4,9]$ | one parameter for PTA |
| **scalar induced GW Background, w/o SMBHBs** | | | |
| $A_\zeta, \Delta = 1$ | Power spectrum amplitude | log-Uniform $[-3,-1.52]$ | one parameter for PTA |
| $A_\zeta, \Delta = 0.05$ | Power spectrum amplitude | log-Uniform $[-3,-1.65]$ | one parameter for PTA |
| $k_\star[{\rm Mpc}^{-1}]$ | Peak scale of the power spectrum | log-Uniform $[4,9]$ | one parameter for PTA |
| **scalar induced GW Background, w/o SMBHBs, w/ HD correlations** | | | |
| $A_\zeta$ | Power spectrum amplitude | log-Uniform $[-3,-1.22]$ | one parameter for PTA |
| $k_\star[{\rm Mpc}^{-1}]$ | Peak scale of the power spectrum | log-Uniform $[4,9]$ | one parameter for PTA |
| $\Delta$ | Width of the power spectrum | log-Uniform $[\log_{10}(0.5),\log_{10} 3]$ | one parameter for PTA |
| **Supermassive Black Hole Binaries (SMBHBs)** | | | |
| $A_{\rm GWB}$ | Strain amplitude | log-Uniform $[-18,-13]$ | one parameter for PTA |

Table 2: List of noise and astrophysical GW background parameters used in our upper limit analyses, together with their prior ranges.

| Parameter | Description | Prior | Comments |
|---|---|---|---|
| **Upper limit analysis** | | | |
| Parameter | Description | Prior | Comments |
| **White Noise** | | | |
| $E_k$ | EFAC per backend/receiver system | Uniform $[0,10]$ | single-pulsar only |
| $Q_k[s]$ | EQUAD per backend/receiver system | log-Uniform $[-8.5,-5]$ | single-pulsar only |
| $J_k[s]$ | ECORR per backend/receiver system | log-Uniform $[-8.5,-5]$ | single-pulsar only (NG12, NG9) |
| **Red Noise** | | | |
| $A_{\rm red}$ | Red noise power-law amplitude | Linear-Exponent $[-20,-11]$ | one parameter per pulsar |
| $\gamma_{\rm red}$ | Red noise power-law spectral index | Uniform $[0,7]$ | one parameter per pulsar |
| **DM Variations Gaussian Process Noise** | | | |
| $A_{\rm DM}$ | DM noise power-law amplitude | Linear-Exponent $[-20,-11]$ | one parameter per pulsar (IPTADR2) |
| $\gamma_{\rm DM}$ | DM noise power-law spectral index | Uniform $[0,7]$ | one parameter per pulsar (IPTADR2) |
| **scalar induced GW Background** | | | |
| $A_\zeta$ | Power spectrum amplitude | Linear-Exponent $[-3,0.]$ | one parameter for PTA |
| $k_\star[{\rm Mpc}^{-1}]$ | Peak scale of the power spectrum | Fixed, see text | one parameter for PTA |
| **Supermassive Black Hole Binaries (SMBHBs)** | | | |
| $A_{\rm SMBHBs}$, NG12 | Strain amplitude | Fixed to $-14.57 (-14.86)$ | one parameter for PTA |
| $A_{\rm SMBHBs}$, IPTA DR2 | Strain amplitude | Fixed to $-14.4 (-14.7)$ | one parameter for PTA |

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
