# Peer review of "Search for scalar induced gravitational waves in the International Pulsar Timing Array Data Release 2 and NANOgrav 12.5 years datasets"

_SciPost Physics, doi:SciPost Phys. Core 6, 060 (2023)_

## Round 1 · Referee Report · Anonymous · 2023-6-29

Strengths
This paper compares three previous papers (Refs. [42,43,44]) that explain the NANOGrav12.5yr data by using the induced gravitational wave (IGW). This is good for communiteis to understand the current status of the data of the NANOGrav12.5yr collaboration.
Weaknesses
I think there is no weak point in this article.
Report
This paper is very valuable in that it carefully compares three previous papers (Refs. [42,43,44]) that explain the NANOGrav12.5yr data by using the induced gravitational wave (IGW).
However, before accepting this paper for publication, I would like to make one comment.
When one obtains a conservative limit on the free parameters in the analysis of a beyond the standard model, such as IGW, due to a large curvature perturbation that creates a Primordial Black Hole, the most conservative bound is the one obtained without considering the contribution of other models, i.e., the Astrophysical signal components from the Supermassive Black Holes mergers. When both are considered, the limit on each model parameter is always stronger than when only one of them is considered. This is at odds with the position of getting a CONSERVATIVE limit. I think that is the reason why the previous three papers did not
consider possible contributions from such unknown astrophysical sources.
Requested changes
None
Author: Fabrizio Rompineve on 2023-07-03 [id 3777]
(in reply to Report 1 on 2023-06-29)
We thank the referee for the useful comment. We have added a clarification in our manuscript, at the end of the third paragraph on p. 10, mentioning the alternative strategy to set upper limits suggested by the referee and motivating our different strategy.
We agree that the weakest upper limits from PTAs on any gravitational wave (GW) signal are obtained by requiring that the latter is not stronger than the currently reported excess, in the absence of any other contribution to the stochastic background. Our different strategy in this paper is motivated by the cosmological and astrophysical constraints on scalar-induced GWs, which very significantly constrain the possibility that scalar-induced GWs can provide a feasible model of the currently reported common process, as shown in our Figs. 3 and 6 and as discussed extensively in Sec. IV of our paper.
Anonymous on 2023-06-28 [id 3772]
This paper is very valuable in that it carefully compares three previous papers (Refs. [42,43,44]) that explain the NANOGrav12.5yr data by using the induced gravitational wave (IGW).
However, before accepting this paper for publication, I would like to make one comment.
When one obtains a conservative limit on the free parameters in the analysis of a beyond the standard model, such as IGW, due to a large curvature perturbation that creates a Primordial Black Hole, the most conservative bound is the one obtained without considering the contribution of other models, i.e., the Astrophysical signal components from the Supermassive Black Holes mergers. When both are considered, the limit on each model parameter is always stronger than when only one of them is considered. This is at odds with the position of getting a CONSERVATIVE limit. I think that is the reason why the previous three papers did not
consider possible contributions from such unknown astrophysical sources.

---

## Round 2 · Referee Report · Anonymous (Referee 1) · 2023-7-4

Strengths

This is a very timely paper to understand the observational possibilities of
induced gravitational wave.

Weaknesses

None

Report

The authors answered all of my questions. I recommend this paper to be published in this journal.

---

## Round 2 · Author Response

Dear Editor,

we thank you and the referee for the report.

We are resubmitting our manuscript following the referee report. We describe below the minor changes done to address the referee's point. We reply to referee in the comment section.

---

## Round 2 · List of Changes

1. We have added a clarification in our manuscript, at the end of the third paragraph on p. 10 (from "An alternative strategy" to "PTA data", mentioning the alternative strategy to set upper limits suggested by the referee and motivating our different strategy.

  2. We have added a new reference in the caption of Fig. 6 (ref. 87 of the new version).

---

## Editorial Decision

published